# Sprouts of *Moringa oleifera* Lam.: Germination, Polyphenol Content and Antioxidant Activity

**DOI:** 10.3390/molecules27248774

**Published:** 2022-12-10

**Authors:** Martina Cirlini, Lorenzo Del Vecchio, Leandra Leto, Federica Russo, Luca Dellafiora, Valeria Guarrasi, Benedetta Chiancone

**Affiliations:** 1Department of Food and Drug, University of Parma, Viale Parco Area delle Scienze 27/A, 43124 Parma, Italy; 2Institute of Biophysics, National Research Council (CNR), Via Ugo La Malfa 153, 90146 Palermo, Italy

**Keywords:** antioxidant activity, bioactive compounds, gibberellic acid, in vitro tissue culture, Moringa, seed germination, sprout characterization

## Abstract

(1) Background: In recent years, the consumption of sprouts, thanks to their high nutritional value, and the presence of bioactive compounds with antioxidant, antiviral and antibacterial properties, is becoming an increasingly widespread habit. *Moringa oleifera* Lam. (Moringa) seems to be an inexhaustible resource considering that many parts may be used as food or in traditional medicine; on the other hand, Moringa sprouts still lack a proper characterization needing further insights to envisage novel uses and applications. (2) Methods: In this study, a rapid and easy protocol to induce the in vivo and in vitro germination of Moringa seeds has been set up to obtain sprouts and cotyledons to be evaluated for their chemical composition. Moreover, the effects of sprouts developmental stage, type of sowing substrate, and gibberellic acid use on the chemical characteristics of extracts have been evaluated. (3) Results: Moringa seeds have a high germinability, both in in vivo and in vitro conditions. In addition, the extracts obtained have different total phenolic content and antioxidant activity. (4) Conclusions: This research provides a first-line evidence to evaluate Moringa sprouts as future novel functional food or as a valuable source of bioactive compounds.

## 1. Introduction

*Moringa oleifera* Lam. (Moringa) is a perennial tree native to India that spread in the tropical and sub-tropical regions [1,2,3], such as Philippines, Cambodia, Central America, North and South America and the Caribbean Islands [4]. Cultivations of Moringa are reported also in Sicily, an island in Southern Italy (personal communication). In the areas of origin, Moringa plays a key role in human and animal nutrition, since its leaves, roots, flowers, fruits and seeds are edible. In particular, leaves are rich in nutrients and represent an important resource, being available at the end of the dry season when access to food may be limited [5,6,7,8,9]. Also, seeds are a versatile food which are consumed green, roasted, powdered, steeped for tea or used to extract oil [10]. Besides its important nutritional role, *M. oleifera* L. has been exploited for its medicinal properties. Indeed, Moringa leaves are rich in β-carotene, proteins, vitamin C, calcium and potassium and act as a good source of natural antioxidants [11,12,13,14,15]. Moreover, in traditional medicine, Moringa has been used to stimulate woman’s milk production and to treat anaemia [15,16].

Over the last decades, research community has challenged the most popular beliefs about Moringa emphasizing its possible roles in medical, phytochemical and pharmacological fields, besides the well-known nutritional properties of leaves and seeds. In this respect, a high content of bioactive compounds relevant to the fields mentioned above has been described in leaves, flowers, roots and bark including but not limited to antioxidants, ascorbic acid, flavonoids, phenolic compounds, carotenoids [1,5], mineral salts, vitamins (vitamins A, B and C, α-tocopherol, riboflavin, nicotinic acid, folic acid, pyridoxine, β-carotene) and essential amino acids (i.e., methionine, cystine, tryptophan and lysine). The presence of these compounds confirms the importance of using Moringa as a valuable dietary supplement [1] with hypotensive [17], potential anti-tumor [18,19] and hepatoprotective activity [20]. Nevertheless, despite the wide number of evidence about the health and nutritional properties of Moringa, sprouts are still poorly characterized for their chemical composition and, subsequently, for their nutritional and medical value [21,22]. Indeed, although sprouts are part of the traditional diet of Japan, Thailand and other Far-Eastern countries, they are being introduced into the diets worldwide. Actually, during the sprouting, many nutrients present in the seeds, undergo transformations that increase their biological value. Moreover, studies carried out on sprouts of different species, legumes, cereals and vegetables, demonstrated that they produce bioactive compounds with antioxidant, antiviral and antibacterial properties [23].

This study aims at filling the gap of knowledge of chemical composition of Moringa sprout, specifically in terms of polyphenol content and antioxidant activity. Traditionally, Moringa is propagated by seed thanks to the high rate and rapidity of germination (ranging between 60% and 90% for fresh seeds, and between 7 to 30 days after sowing, respectively) [24,25]. Unfortunately, like many tropical seeds, Moringa seeds loose viability quickly, potentially due to their high metabolic rate [26] and their high oil content [25]. To exploit the potential of Moringa sprouts, it is necessary to study the germination process, in conditions not influenced by environment and by the season. In this respect, the in vitro culture represents a valuable framework of analysis. Indeed, several studies reported that Moringa seeds respond well to in vitro conditions showing a time course of germination and growth faster than in greenhouse conditions [12,27]. Moreover, in vitro tissue cultures have many advantages including the independence from climatic and geographical conditions, reduced space requirements and faster plant growth. This may ensure a continuous and sustainable production of sprouts meant to be used as such, or for secondary metabolites production [28,29]. Finally, tissue culture techniques may also allow modulating the production of bioactive compounds acting on the composition of culture medium in terms of growth regulators content [30,31]. Several studies investigated and confirmed the specific influence of gibberellic acid (GA_3_) as elicitor for triggering secondary metabolism [32,33]. An addition of GA_3_ in the culture medium may result in a significant accumulation of caftaric and cichoric acid in *Echinacea* roots as stated by Abbasi et al. [34], an increased accumulation of cumarin in *C. intybus* roots as reported by Bais et al. [35], and a positive influence on the synthesis of Artemisinin in hairy roots of *Artemisia annua* [36,37]. This clearly demonstrates that the Ga_3_ concentration may represent a key factor to modulate secondary metabolism [37].

To the best of our knowledge, this study thoroughly evaluated for the first time polyphenols and antioxidants synthesis in Moringa sprouts and cotyledons under different conditions such as diverse growing environments (in vivo and in vitro) and substrates (Jiffy^®^ and agarized medium), developmental stages and content of GA_3_ in the culture medium.

## 2. Results

### 2.1. Moringa Seed Germination

#### 2.1.1. In Vivo Seed Sowing

Moringa seed germination is hypogeal, in fact, the cotyledons remain beneath the soil surface, where they have been deposited [38]. For this reason, the first sign of emergence was the epicotyl carrying the first true leaves. The first seed emergence was observed around seven days after sowing; after that, seedlings kept emerging for 32 days, reaching the Final Emergence Percentage (FEP) of 89% (Yellow curve in Figure 1). Mean Emergence Time (MET), calculated to evaluate the average time seeds took to germinate, showed that on average, seeds germinated after 14.6 days. Moreover, the calculated Germination Value (GV) (13.10%) and Germination Index (GI) (1.96 germinated seeds per day) evidenced that, when sown in vivo, Moringa seeds germinated slowly and asynchronously (Appendix A) 

#### 2.1.2. In Vitro Seed Germination

Monitoring, performed on in vitro cultured seeds, carried out for 22 days, enabled the observation of the whole Moringa seed germination process (Figure 2). Seeds swelled promptly and the protrusion of the first radicle was observed after less than 48 h of culture. In 4 days, around 33% of cultured seeds germinated, independently on the culture medium composition (Figure 1). After the first root appearance, germinated seeds, moved to jars, kept growing, following the normal seed development (Figure 3); in 9 days, around 90% of sprouts had reached the stage 2 development, with the emergence of the epicotyl and the first true leaves. 

#### 2.1.3. Effect of Culture Medium Composition on In Vitro Moringa Seed Germination

The influence of the factor “Culture medium composition” on descriptive parameters of Moringa seed germination was analyzed statistically but no significant differences among conditions were seen. At the end of the experiment, an average of 96% of Final Germination Percentage (FGP) was observed and it seemed that adding GA_3_ could not induce seeds to germinate further (Figure 4). The lack of effect related with the presence of GA_3_ in the culture medium was also highlighted by the trend of the germinative process as no significant difference was observed for any of the monitoring dates (data not shown).

The presence of GA_3_ in the culture medium did not influence the Mean Germination Time (MGT); Moringa seeds needed, on the average 6.4 days, to germinate, independently of the culture medium tested (Figure 4). 

Considering the parameter GV that gives an indication of the rapidity of germination, results confirmed that the addition of GA_3_ in the culture medium is ineffective to the germination process (Appendix A) and the average GV recorded was 56.9%. The same trend was observed for the parameter measuring the percentage and speed of germination: in this study, GI was not influenced by the factor “Culture medium composition” (Appendix A), and an average GI of 2.74 seeds per day was registered. Finally, it is worth mentioning that for all the parameters evaluated, Moringa seeds in vitro cultured showed FGP higher than those in vivo sown, regardless the culture medium composition (96% vs. 89%). Moreover, in vitro cultured seeds needed a shorter time (6.4 vs. 14.6 days) to germinate and the germination process was much more synchronous (56.9% vs. 13.1%), confirming the validity of in vitro culture techniques for large-scale Moringa sprout production.

### 2.2. Chemical Characterization of Moringa sprouts

#### 2.2.1. Effect of Developmental Stage and Type of Sowing Substrate on the Total Polyphenol Content and Antioxidant Activity of Moringa sprout Extracts

Extracts, obtained from Moringa sprouts, collected at two developmental stages, two and four leaves, and grown in a growth chamber, in glass jars on agarized HF medium and in Jiffy^®^ pots, were chemically characterized, to evaluate their total polyphenol content (TPC) and antioxidant activity (Table 1).

Statistical analysis carried out on data recovered from the tests, evidenced a strong influence of both factors, “Sprout Developmental Stage” and “Type of Sowing Substrate” on the extract composition. Specifically, the results of Folin-Ciocalteau analysis, performed to evaluate the TPC in the Moringa sprout extracts, revealed that the “Type of Sowing Substrate” was the only effective factor. Extracts obtained from in vitro grown sprouts were indeed statistically richer in polyphenols than those from in vivo grown sprouts (119.71 ± 11.19 mg GAE/g vs. 76.35 ± 23.14 mg GAE/g).

To attest the antioxidant activity of extracts, several chemical assays were carried out as detailed below. The first was the DPPH assay and statistical analysis evidenced that the sprout antioxidant activity was influenced by the interaction of the two factors considered, with a predominant effect resulting from the “Type of Sowing Substrate” factor. Effectively, extracts from stage 1 in vitro cultured sprouts were characterized by an antioxidant activity statistically higher than those, grown in Jiffy^®^ at the same developmental stage (190.90 ± 12.47 mM TEAC vs. 48.37 ± 16.35 mM TEAC). The antioxidant activity measured with ABTS assay appeared to be exclusively determined by the “Type of Sowing Substrate”, with extracts from in vitro grown sprouts showing higher values than those grown in Jiffy^®^ pots (185.14 ± 14.25 mM TEAC vs. 122.81 ± 17.38 mM TEAC) (Table 1). Finally, a different response was recorded with the results of the FRAP assay, where a significant interaction between the two factors was observed. These results highlighted that extracts from stage 1 in vitro grown sprouts had an antioxidant activity higher than those at stage 2 (134.56 ± 14.49 vs. 72.47 ± 0.51). On the other hand, in in vivo the behaviour was opposite and extracts from stage 2 sprouts have a higher activity than those obtained from stage 1 (78.25 ± 0.19 mM TEAC vs. 7.72 ± 2.02 mM TEAC). 

#### 2.2.2. Effect of Developmental Stage and Type of Sowing Substrate on the Total Polyphenol Content and Antioxidant Activity of Moringa Cotyledon Extracts

Together with the sprouts, the cotyledons were also chemically analyzed to determine how the developmental stage of the sprout and the type of substrate may influence their polyphenol content and their antioxidant activity (Table 2). Particularly, regarding the parameter TPC, “Type of Sowing Substrate” was the only significant factor, with cotyledons grown in Jiffy^®^ showing a polyphenol content statistically higher than those observed in cotyledons grown in vitro (9.98 ± 2.93 mg GAE/g vs. 4.29 ± 1.45 mg GAE/g).

A significant interaction was observed also for the DPPH assay. Indeed, within sprouts grown in Jiffy^®^, extracts from stage 2 cotyledons showed a much higher antioxidant activity than those from stage 1 (36.12 ± 1.55 mM TEAC vs. 7.02 ± 1.60 mM TEAC). Furthermore, an opposite trend was observed considering the influence of the factor “Type of Sowing Substrate” as an intense antioxidant activity was recorded for extracts from stage 1 cotyledons, when sprouts grew in vitro (25.92 ± 2.22 mM TEAC vs. 23.52 ± 0.82 mM TEAC), and from stage 2, when sprouts grew in Jiffy^®^ (36.12 ± 1.55 mM TEAC vs. 7.02 ± 1.60 mM TEAC) (Table 2). 

Results for the ABTS assay evidenced that the factor “Type of Sowing Substrate” was the main source of variation (Table 2); indeed, extracts from cotyledons of sprouts grown in Jiffy^®^ showed an antioxidant activity higher than those cultured in vitro (41.38 ± 0.14 mM TEAC vs. 32.01 ± 8.56 mM TEAC). 

Finally, for FRAP assay a significant interaction between factor was recorded and it seems that, within extracts from stage 1 cotyledons, those from in vitro cultured sprouts had a higher antioxidant activity than the ones grown in Jiffy^®^ (11.61 ± 0.25 mM TEAC vs. 2.22 ± 0.78 mM TEAC). An opposite trend was observed for extracts from stage 2 cotyledons as a statistical higher activity was recorded in extracts from sprouts grown in Jiffy^®^. Moreover, if the substrate is considered, it appeared that extracts from stage 2 cotyledons, grown in Jiffy^®^, had a higher antioxidant activity than those from stage 1 (14.36 ± 0.36 mM TEAC vs. 2.22 ± 0.78 mM TEAC) (Table 2).

#### 2.2.3. Effect of Developmental Stage and Culture Medium Composition on Total Polyphenol Content and Antioxidant Activity of Moringa sprout Extracts

The statistical analysis carried on the results obtained from the chemical analysis of extracts from Moringa in vitro cultured sprouts exhibited a significant influence of both tested factors “Sprout Developmental Stage” and “Culture Medium Composition” on the polyphenol content and antioxidant activity (Table 3).

The polyphenol content of extracts, measured using Folin-Ciocalteau assay, seemed to be strongly influenced by the factor “Sprout Developmental Stage”. Indeed, extracts from stage 1 sprouts resulted, on average, richer in polyphenols than extracts from stage 2 sprouts (86.55 ± 22.77 mg GAE/g vs. 164.55 ± 63.82 mg GAE/g), regardless of the culture medium composition. The measure of antioxidant activity through DPPH, ABTS and FRAP assays showed a significant interaction between the factors tested suggesting that the presence of GA_3_ in the culture medium significantly increased the antioxidant activity of extracts, specifically from stage 2 (Table 3). 

#### 2.2.4. Effect of Developmental Stage and Culture Medium Composition on the Polyphenol Content and Antioxidant Activity of Moringa cotyledon Extracts

The chemical analysis carried out on the extracts obtained from cotyledons, belonging from in vitro cultured sprouts, detected a strong influence of both factors considered, “Sprout Developmental Stage” and “Culture Medium Composition” (Table 4). For TPC, DPPH and FRAP assays, the interaction of both factors exerted a strong influence. Specifically, in extracts from stage2 cotyledons, the addition of GA_3_ in the culture medium appeared to increase the polyphenol content. Conversely, the statistical analysis carried on the ABTS assay results evidenced that the main influence on antioxidant activity of extracts was the “Sprout Developmental Stage” factor, with higher values on average recorded in extracts from stage 2 cotyledons (37.78 ± 3.29 mg GAE/g vs. 26.38 ± 1.47 mg GAE/g). 

## 3. Discussion

Moringa is a crop that has drawn a growing attention over the last decades thanks to its multifaceted properties that make this plant suitable not only for human and animal nutrition, but also for medical, pharmaceutical and cosmetical purposes. Leaves, flowers, stems, seeds and roots are the Moringa’s parts most widely studied, and they have been used as food, feed and applied in traditional medicine [39,40]. The promising characteristics have been scientifically confirmed by several studies reporting the Moringa health-promoting properties [41,42]. 

The well-documented health-promoting potential of Moringa lead to an increase in its cultivation. However, unfortunately, direct seeding, which is among the most adopted propagation methods on account for its easy applicability and cost-effectiveness [43], presents some flaws, which are mostly related to fast decrease of seed viability and slow plant growing [44,45,46,47,48]. A strategy to solve this kind of problems is through plant biotechnology. Several studies explored alternative propagation methods through in vitro tissue culture [12,40,48,49] although very few dealt with in vitro seed germination [50]. This dearth of data made it difficult to perform comparative discussion of our results. However, based on the few data available, the germinability observed in vitro and in vivo was comparable with that reported in literature. Usually, Moringa seeds have a high germination rate, ranging from 60% to 100% [24]. In the present study, comparable values were observed being 89% of seeds germinated when sown in Jiffy, and an average of 96% when cultured in vitro. These results confirm that in vitro culture techniques can replace traditional sowing methods, particularly when the environmental conditions are not ideal. Nevertheless, although in vivo and in vitro sowing methods comparably promote seeds germination, the in vitro culture of Moringa seeds seemed to have better performances on account for the results obtained on MGT. Indeed, this research reports that an average of 14 days is needed for seeds to germinate when cultured in vivo, and 6.4 days for seeds cultured in vitro, with a huge effect to reduce the time to obtain well developed sprouts. Results were in agreement with data from the literature describing that, depending on the culture conditions and the seed treatments, the MGT ranges from 13.5 days to 17.7 [51,52]. 

The role of GA_3_ added to the in vitro culture medium deserves a particular mention as it was never duly investigated before to the best of our knowledge. GA_3_ is a well-known growth regulator, designated to break seed dormancy and to promote germination in several plant species, Moringa included [53]. Several studies reported that the seed imbibition in solutions containing GA_3_ as pre-treatment may enhance in vivo germination [51]. However, in this study, no significant differences in the in vitro cultured seed germinability were observed. 

Moringa is considered a multipurpose plant, with various parts consumed as food and used in traditional medicine [54], widely studied for the sake to deepen the understanding of health-promoting properties [55,56]. Even if the consumption of sprouts of several plant species has spread all over the world, few are studies investigating the biological value of Moringa sprouts, specifically considering the growing conditions [21,22]. In this study, an efficient germination protocol was obtained first. Then, Moringa sprouts were characterized for their chemical composition considering the use of different growing substrate, the sprout developmental stage, and, only for those obtained in vitro, the presence of GA_3_ in the culture medium. Specifically, results detected in this study, in terms of sprout TPC, turned out to be way more interesting and encouraging than those reported in literature for other Moringa organs. As an example, da Silva et al., in 2020 [57], investigated the phenolic composition of different organs of *Moringa oleifera*, as leaves, hypocotyls and roots, and reported values ranging between 2.185 ± 0.089 mg GAE/g and 3.805 ± 0.304 mg GAE/g. Of note, these values are considerably lower than those obtained from the Moringa sprouts analyzed in this study (58.29 ± 0.66–123.73 ± 16.43 mg GAE/g, Table 1). These considerably higher TPC values found in sprouts agreed with the data from the literature being the phenolic content highly variable depending on the different organs considered [57,58]. TPC of sprouts and cotyledons reported in this study are very different with a TPC value found in sprout samples fourteen times higher than in the cotyledons, independently of the developmental stage and growing substrate considered. Moreover, same authors reported that the biosynthesis and accumulation of bioactive substances may depend on the environmental conditions the plants are subjected to during cultivation and harvesting [57,58]. In this study, the gap between TPC of *vitro* and *vivo*-cultured sprouts is huge (on the average, respectively, 119.70 ± 5.69 mg GAE/g and 76.35 ± 25.54 mg GAE/g). Besides the already cited factors affecting the TPC in Moringa, also the developmental stage of the different organs analyzed seems to have a crucial effect. In particular, Sreelatha and Padma [59] demonstrated that Moringa leaves presented a phenol quantity of 36.02 ± 0.01–45.81 ± 0.02 mg GAE/g, depending on the maturity of analyzed leaves. Also in this study, the developmental stage of the analyzed sprouts plays a key role, with the TPC increasing as the sprout develop from stage 1 to stage 2. A more recent study conducted on Moringa leaves stated that the TPC values obtained for the leaf’s extracts ranged between 21.7 ± 1.6 mg GAE/g and 39.1 ± 3.3 mg GAE/g, regardless of the extraction conditions used [60]. Therefore, the high TPC values obtained from Moringa sprouts analyzed in this study are encouraging, also considering the limited effects the methods of extraction may have. 

Besides TPC, Moringa sprouts and cotyledons were tested for the antioxidant activity of their extracts via three different methodologies, namely DPPH, ABTS and FRAP assays. Results indicated that the antioxidant properties of the extracts reflected their phenolic composition. As reported for TPC, the antioxidant capacity may vary depending on the type of explant considered. In more detail, extracts obtained from sprouts showed an antioxidant activity around 10 times higher than those from cotyledons, independently of developmental stage and growing substrate of explants. The same trend is confirmed by Santos et al. [61] who tested through DPPH radical scavenging activity test saline and alcoholic extracts obtained from several Moringa plant organs (i.e., leaves, rachis, stem, flowers and seeds) demonstrating that the antioxidant capacity of the extracts markedly depends on the plant organ. The organ-dependency of antioxidant activity of Moringa extract is also reported by Gómez-Martínez et al. [58], who tested the antioxidant power of Moringa leaflets and petioles via DPPH, ABTS, and FRAP assays. Furthermore, in the endosperm or in the cotyledons of the embryo, seeds can accumulate secondary metabolites that will spread in the sprout during its development [62]. The reassortment and the ex novo synthesis of secondary metabolites are highly influenced by several factors and they are both typically susceptible to external stimuli [63]. Thus, the diverse culture systems applied in this study, the different types of matrices (sprouts and cotyledons) and substrates (agarized medium and Jiffy^®^) are likely to have triggered different secondary metabolite syntheses.

Exogenous phytohormones are routinely used in in vitro culture not only to regulate the growth and differentiation of plant cells, tissues, and organs, but also as elicitors, which are referred to as agents stimulating secondary metabolism in plants to synthesize important compounds, including but not limited to anthocyanins and flavonoids [34,36,64,65]. Nowadays, the mechanisms elicitors adopt to interact with secondary metabolism are still unclear, but it is plausible that in vitro cultured plant tissues react to the presence of an elicitor as the plants react to pathogens, activating defence mechanism that stimulate the synthesis of secondary metabolites [66]. Among the different classes of phytohormones used as elicitors, gibberellins, and, specifically, GA_3_ has been used to stimulate the production of secondary metabolites, such as phenols [33] and tanshinones in *S. miltiorrhiza* hairy roots [32], caffeic acid derivatives (CADs) in *Echinacea pupurea* hairy roots [34] and artemisinin in *Artemisia annua* [37,67]. In this study, adding GA_3_ to the culture medium did not influence the TPC in sprouts, but stimulated the synthesis of non-polyphenolic compounds with a strong antioxidant activity. Evidently, the influence of GA_3_ in culture medium greatly varies depending on the starting material analyzed. Indeed, even if GA_3_ did not increase the level of antioxidant compounds in Moringa sprouts, it had an appreciable effect in Moringa cotyledons. In the research reported by Radić et al. [68], where the influence of different combinations of growth regulators was tested to evaluate the total polyphenolics and free radical scavenging capability of *Stevia rebaudiana* leaf, callus and root extracts, the authors demonstrated that each type of explant responds differently to the same culture conditions. Moreover, besides the type of explant, also the GA_3_ concentration seems to influence the secondary metabolism response. For example, in *Stevia rebaudiana* (Bert), the TPC and the antioxidant activity were differentially influenced when GA_3_ was added alone or in combination with other plant hormones [68]. Conversely, GA_3_ induced a considerable increase in free radical activity of *Echinacea purpurea* hairy roots extracts when added in low concentration [34]. In this study, the presence of GA_3_ alone, increased the antioxidant activity of both sprout and cotyledon extracts in a concentration-independent manner. These results are similar to those reported by Ali et al. [69], who tested three concentrations of GA_3_ (0.5, 1.0, 2.0 mg/L) and observed a positive correlation between TPC and the presence of GA_3_ in treated cell suspension cultures of *Artemisia absinthium* L.

## 4. Materials and Methods

### 4.1. Plant Material

Healthy uniform Moringa seeds, harvested in December from trees grown in Sicily, an island in Southern Italy, were kindly provided by a local grower. Seeds had an average mass of 0.66 g with outer teguments (Figure 5a) and 0.46 g without (Figure 5b), their average size were, respectively, of 12.0 × 15.0 mm and 10.0 × 10.0 mm. They were stored in paper bags at room temperature and darkness for 6 months until their use. 

### 4.2. Moringa Seed Germination

#### 4.2.1. In Vivo Seed Sowing

Moringa seeds were deprived of their pods and sown in Jiffy^®^, previously re-hydrated with sterile distilled water. 50 seeds were sown (2 cm deep) in hydrated Jiffy^®^ and stored in a growth chamber, at 25 ± 1 °C and light intensity of 20 μmol m^−2^ s^−1^, under 16 h photoperiod. 

#### 4.2.2. In Vitro Establishment of Culture

To reduce the risk of contamination and ameliorate the germination process, outer teguments of Moringa seeds were removed, while the inner part was kept to protect teguments during the sterilization procedure. Seed surface sterilization was carried out, inside a laminar flow cabinet, by immersion in 70% ethanol (*v*/*v*) for 5 min and 25% commercial bleach (*v*/*v*) for 20 min, followed by rinsing three times in sterile distilled water. Sterilized seeds were placed in Petri dishes (100 × 15 mm) containing 10 mL of HF culture medium. HF culture medium composition was the following: Murashige and Skoog (MS) basal salt mixture (1×) [33], MS vitamin mixture (1×) [70], 30.0 g/L of sucrose and 8.0 g/L of Agar. HF culture medium, after adjusting the pH to 5.8 with 1N NaOH, was sterilized in autoclave for 20 min at 121 °C. Germinated seeds were moved to 500 mL glass jars with 100 mL of HF culture medium, in order to allow a better sprout development. Petri dishes and glass jars were sealed and maintained in a growth chamber, at 25 ± 1 °C and light intensity of 20 μmol m^−2^ s^−1^, under 16 h photoperiod. 

#### 4.2.3. Effect of Culture Medium Composition on In Vitro Moringa Seed Germination

To increase Moringa germinability, the influence of two concentrations of Gibberellic Acid (GA_3_) added to the culture medium was evaluated; specifically, three culture media with the following composition were tested: (1) 0.0 G: HF medium; (2) 0.5 G: HF added with 0.5 mg/L of GA_3_ and (3) 1.0 G: HF added with 1.0 mg/L of GA_3_. Per each culture medium, 50 seeds were cultured, 5 seeds per each Petri dish (100 × 15 mm), each containing 10 mL of culture medium (10 Petri dishes per treatment). 

#### 4.2.4. Analysis of Data

Seeds were monitored every two days for 22 days, for those cultured in vitro, and for 32 days for those sown in Jiffy^®^; after these dates, no more germinative events were observed. Germination curves were used to illustrate the germination test results.

Data recorded were used to measure the following different descriptive parameters: Emergence Percentage (EP) and Mean Emergence Time (MET) for the in vivo cultured seeds and Final Germination Percentage (FGP) and Mean Germination Time (MGT) for the in vitro cultured seeds. Moreover, for both in vivo and in vitro cultured seeds, Germination value (GV) and Germination Index (GI) were calculated; the formulae used were the following: EP = 100 × ng/ft (ng is the number of emerged seeds and ft is the total number of cultured seeds); FGP = 100 × fg/ft (fg is the number of germinated seeds and ft is the total number of cultured seeds); MET = ∑n × x/∑f (n = seeds emerged, x = n° of days from seeding corresponding at day x) [71]; MGT = ∑ƒ × x/∑ƒ (ƒ = seeds germinated, x = n° of days from seeding corresponding at day x) [72]; GV= Peak Value (PV) * Mean Daily Germination (MDG) (Peak Value: maximum quotient derived from all of the cumulative germinated seed percentages on any day divided by the number of days to reach this percentages) [73]; MDG: mean daily germination calculated as the percentage of full-sprouted nodal explant at the end of the test divided by the number of days to the end of the test); GI = Σ(fx/Di) (f = number of germinated seeds at day ‘x’ and Di is day ‘x’ [74]. One-way ANOVA was used to calculate differences within the factor tested (Culture Medium Composition) per each parameter considered; Tukey’s test (*p* ≤ 0.05) was used for mean separation (SYSTAT 13.1, Systat Software, Inc.; Pint Richmond, CA, USA).

### 4.3. Chemical Characterization of Moringa sprouts and Cotyledons

#### 4.3.1. Plant Material

For chemical characterization, the influence of the culture medium composition and of the type of sowing substrate on Total Phenol Content (TPC) and on the antioxidant activity, sprouts, obtained in vivo (Jiffy^®^ pots) and in vitro and, were analyzed at two developmental stages: when sprouts showed the first two true leaves (Stage 1) (Figure 6a) and when at least four leaves were formed (Stage 2) (Figure 6b). Moreover, for each of the sowing method and developmental stage considered, also the cotyledons (arrows in Figure 6) were subjected to the same analysis of the sprouts. Per each type of explant, sprouts and cotyledons, 1 g of sample was used for the chemical analysis. 

#### 4.3.2. Chemical Materials

Ethanol used for the extraction of compounds of interest was purchased from Carlo Erba (Milan, Italy), while bi-distilled water was obtained from VWR (Milano, Italy). The analytical standards of gallic acid, (±)-6-hydroxy-2,5,7,8-tetramethylchromane-2-carboxylic acid (Trolox), the reagents and salts as 2,2-diphenyl-1-pirylhydrazyl free radical (DPPH), 2,2-azinobis(3-ethylbenzothiazoline-6-sulfonic acid) (ABTS) diammonium salt, 2, 4, 6-tripyridyl-s-triazine (TPTZ), sodium carbonate, potassium persulfate, sodium acetate, acetic acid, hydrochloric acid, and ferric chloride hexahydrate were all purchased from Sigma-Aldrich (St. Louis, MO, USA), while Folin-Ciocalteau’s phenol reagent solution was purchased from VWR (Milano, Italy). 

#### 4.3.3. Extraction of Antioxidant Components

Prior to the extraction step, all the samples considered in this study were subjected to a lyophilisation process for 48 h using a Lio-5P Freeze dryer (5Pascal, Milan, Italy); then all the dried samples were reduced in powder using a laboratory miller. The extraction was conducted as described by Gómez-Martínez et al. [58] with some modifications. In brief, 0.5 g of milled sample were treated with 8ml of Ethanol/bi-distilled water (50/50 *v*/*v*) and extracted on a HS 501 digital shaker (IKA-Werke, Staufen, Germany) at 200 strokes/min for 30 min at room temperature. After that, extracts were centrifuged using a Centrifugette 4206 centrifuge (Alc International, Pévy, France) at room temperature, for 10 min at 5000 rpm. Supernatants were recovered and stored at −20 °C until the analyses.

#### 4.3.4. Determination of Total Polyphenolic Content

TPC was determined on the basis of the protocol reported by da Silva et al. (2020) [57], with slight modifications applied. 250 µL of sample extracts were added with 1 mL of Folin-Ciocalteau’s phenol reagent solution previously, diluted in bi-distilled water (1/10, *v*/*v*) and with 2 mL of aqueous sodium carbonate (20 %, *w*/*v*). The reaction was performed in the dark for 30 min, then all the samples were analyzed on a JASCO V-530 spectrophotometer (Easton, MD, USA), recording absorbance at 760 nm. In order to determine the amount of phenols, a calibration curve based on the measurements of five different gallic acid solutions in the range of 0.01–0.1 mg/g was built, and data obtained for all the samples tested were then expressed as mg/g of gallic acid equivalents (mg GAE/g). More specifically, since all the considered matrices underwent to a lyophilisation process, obtained data are reported on dry matter. 

#### 4.3.5. Evaluation of Antioxidant Activity

The antioxidant activity was evaluated on the basis of three different tests: DPPH radical scavenging activity test

The radical scavenging capacity of the extracts was firstly determined by the DPPH radical scavenging assay, according to Sharma et al. [75]. 100 µL of sample extract were mixed with 2.9 mL of a methanolic DPPH solution (0.05 mM) and kept in the dark at room temperature for 30 min. The absorbance of the samples was, then, recorded at 517 nm using a JASCO V-530 spectrophotometer (Easton, MD, USA). A blank was prepared using 100 µL of extraction solution and then measured, after the incubation, with the DPPH reagent solution. A calibration curve was prepared using Trolox as reference, in a concentration range of 0.1–1 mM (5 points). The radical scavenging capacity was calculated taking into account the percentage of inhibition of the radical. In particular, the following mathematical formula was applied: I% = [(Absblank − Abssample)/Absblank] * 100, were Absblank was the absorbance of the blank sample and Abssample was the absorbance of the standard solution or of sample. Data were reported as TEAC values (Trolox Equivalent Antioxidant Capacity; mM TEAC). All the analyses were conducted in double.
ABTS radical scavenging activity test

ABTS assay was performed following the protocol reported by Gómez-Martínez et al. [58] with some modifications. In particular, a stock aqueous solution of ABTS radical cation (ABTS+) (7 mM) and potassium persulfate (2.45 mM) was prepared and kept in the dark for 16 h, stirring the solution at constant speed. Then, the solution was properly diluted in ethanol to obtain an absorbance of 0.70 ± 0.2 at 734 nm (JASCO V-530 spectrophotometer, Easton, MD, USA), and used for sample analyses. 20 µL of sample extract (or blank or standard solution) were treated with 1.98 mL of ABTS+ diluted solution. The reaction was conducted in the dark at room temperature for 30 min and, after that, the absorbance of all the samples was recorded at 734 nm. The quantification was performed on the basis of Trolox, as described in the case of DPPH test. All the analyses were repeated twice.
Ferric ion reducing power (FRAP)

Finally, the antioxidant capacity of moringa extracts was evaluated also by ferric reducing antioxidant power (FRAP) assay, according to Gómez-Martínez et al. [58]. The FRAP reagent solution was prepared by mixing 2.5 mL of an aqueous solution of TPTZ (10 mM) acidified with hydrochloric acid (40 mM), 2.5 mL of an aqueous solution of ferric chloride hexahydrate (20 mM), and 25 mL of acetate buffer 300 mM (pH = 3.6). The solution was subjected to heating (37 °C) for 30 min before use, then 150 µL of sample extract, blank or Trolox solution, were submitted to the reaction with FRAP solution (2.85 mL) in the dark at room temperature for 30 min, and then the absorbance was recorded at 593 nm (JASCO V-530 spectrophotometer, Easton, MD, USA). The ferric ion reducing activity of samples was estimated on the basis of a Trolox calibration curve (0.1–1 mM, 5 points). Results were expressed as mM TEAC. All the analyses were repeated twice.

### 4.4. Statistical Analysis of Data

Data recovered from the different chemical tests on sprouts and cotyledons, grown in vivo and in vitro, were used to calculate means and two-way ANOVA was carried out considering the influence on the parameters tested (TPC, DPPH, ABTS and FRAP) of the factors “Sprout Developmental Stage” and “Type of Sowing Substrate” for the first set of analysis and “Sprout Developmental Stage” and “Culture Medium Composition” for the second one. Tukey’s test (*p*
< 0.05) was used for mean separation (SYSTAT 13.1, Systat Software, Inc; Pint Richmond, CA, USA).

## 5. Conclusions

The sprouts are traditionally consumed in Eastern cultures, though they have entered diet habits worldwide in the last years also on account of their promising health-promoting effects. It has been indeed demonstrated that the germination process may increase the biological value of many nutrient components present in the seed reducing the content of anti-nutrients. Moreover, it is well recognised that sprouts of several plant species, such as, legumes, cereals and vegetables, contain bioactive compounds with antioxidant, antiviral and antibacterial properties. Moringa has a long history of use as food and as a valuable sourse of medications in traditional medicine, though it is getting attention for it application in the pharmaceutical and nutraceutical sectors. In countries where Moringa is native, many parts of the plants are usually consumed for their numerous nutritional properties, but, to the best of our knowledge, Moringa sprouts were never considered as potential novel food. To assess this potential application of Moringa, it is necessary to set up a valid seed gemination protocol and to carry out a chemical characterization of sprouts as a first-line evidence to support further development in that sense. In this study Moringa seeds were sown in in vivo and in in vitro conditions, testing different types of sowing substrates. Vitro-derived Moringa sprouts are ideal for those interested in a standardized product that can be supplied throughout the year; the higher costs of vitro-derived plant material, due to the necessary equipments and the skilled labour, are balanced by the high and constant quality, independent of the season and of the environmental conditions. Seeds in vitro cultured germinated faster and with a higher percentage of germination, regardless the composition of culture medium composition. The chemical characterization carried out on sprouts and on their cotyledons confirmed these matrices as precious source of bioactive compounds, indicating that they could be applied in different productive sectors. Total polyphenol content and antioxidant capacity of both matrices are remarkable, even with a slight differences between sprouts and cotyledons. Moreover, GA_3_ acts as elicitor stimulating the synthesis of bioactive compounds of Moringa sprouts obtained in vitro, and in their cotyledons, as reported for other species. Even if preliminary, the results reported in this study lay the groundwork to deepen the knowledge to support a scale-up production of Moringa sprouts production and provided a useful chemical characterization for a potential future use of this matrix as food or source of bioactive compounds.

## Figures and Tables

**Figure 1 molecules-27-08774-f001:**
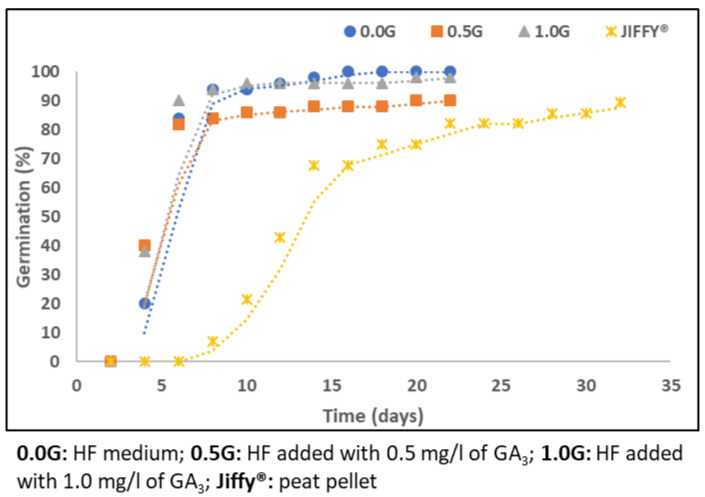
Germination curves of in vivo (Jiffy^®^) and in vitro (agarized Highly Fertile medium) sown Moringa seeds; different curves represent the germination behaviour of samples grown adding increasing concentrations of gibberellic acid (GA3): 0 (0.0 G), 0.5 (0.5 G) and 1 (1.0 G) mg/L, and material cultivated in Jiffy^®^.

**Figure 2 molecules-27-08774-f002:**
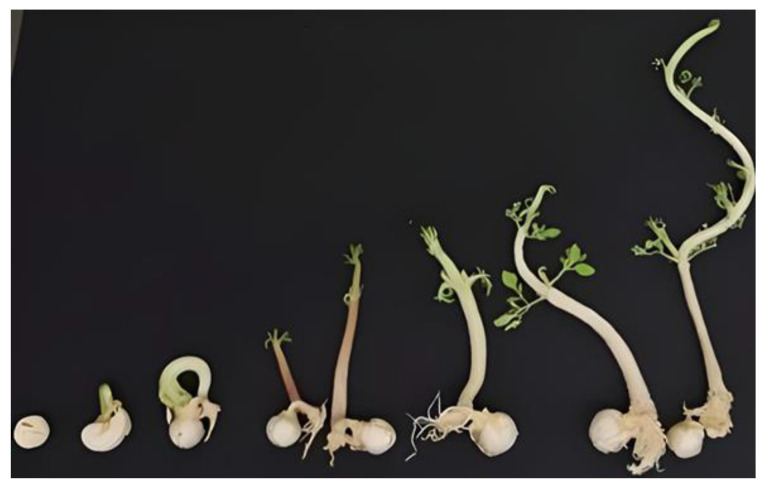
Germinative process of in vitro cultured Moringa seeds.

**Figure 3 molecules-27-08774-f003:**
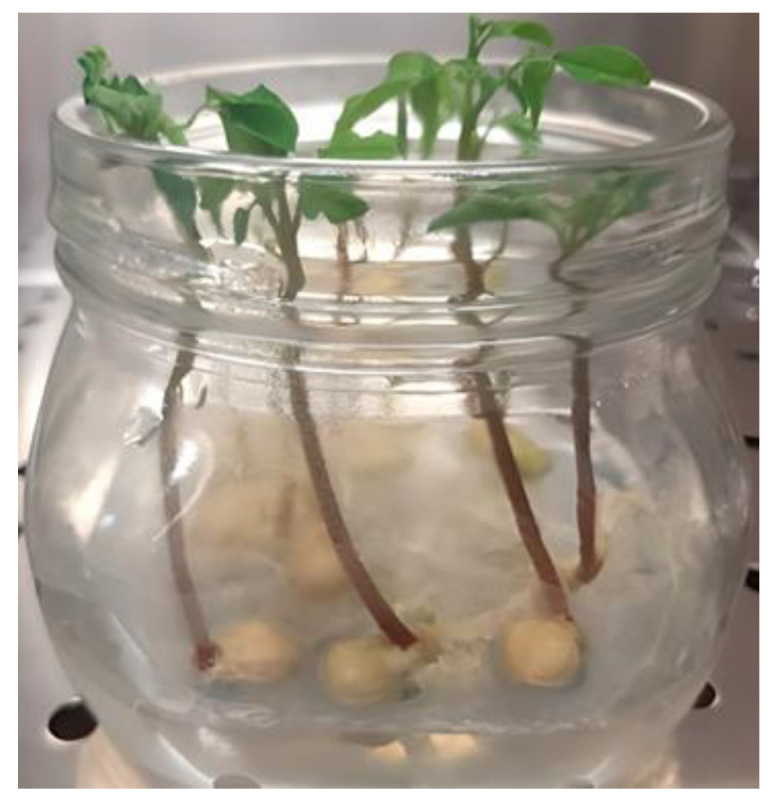
Moringa sprouts growing in a glass jar.

**Figure 4 molecules-27-08774-f004:**
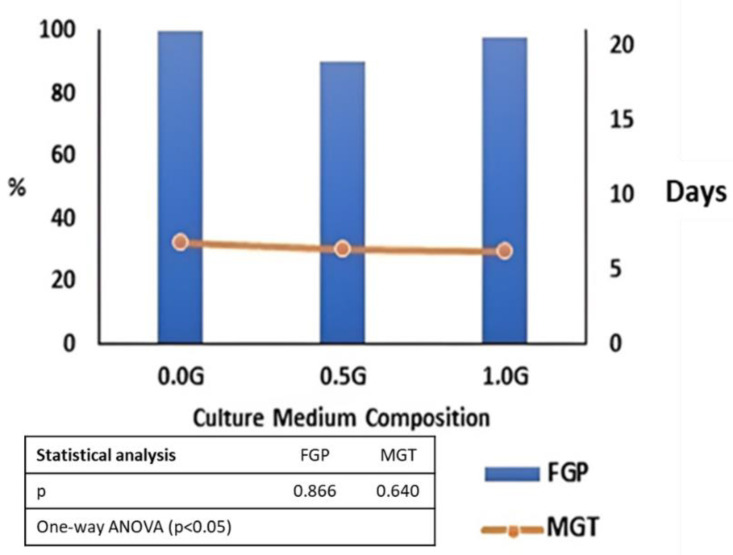
Final Germination Percentage (FGP) and Mean Germination Time (MGT) of in vitro cultured Moringa seeds, sown with increasing concentrations of gibberellic acid (GA_3_): 0 (0.0 G), 0.5 (0.5 G) and 1 (1.0 G) mg/L of GA_3_.

**Figure 5 molecules-27-08774-f005:**
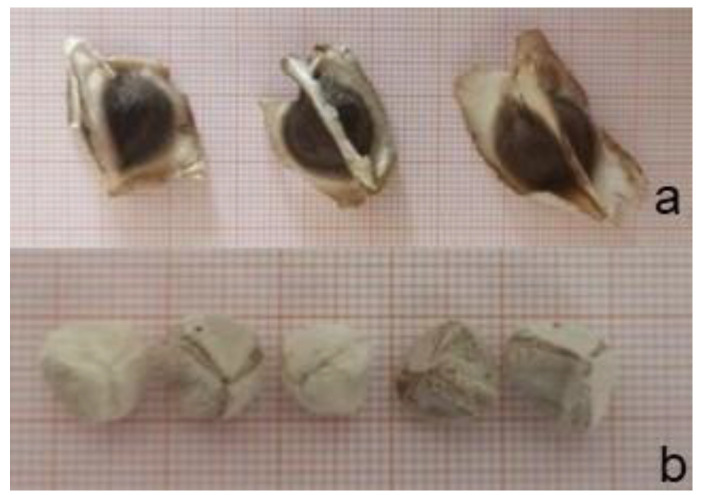
Moringa seeds with (**a**) and without (**b**) outer teguments.

**Figure 6 molecules-27-08774-f006:**
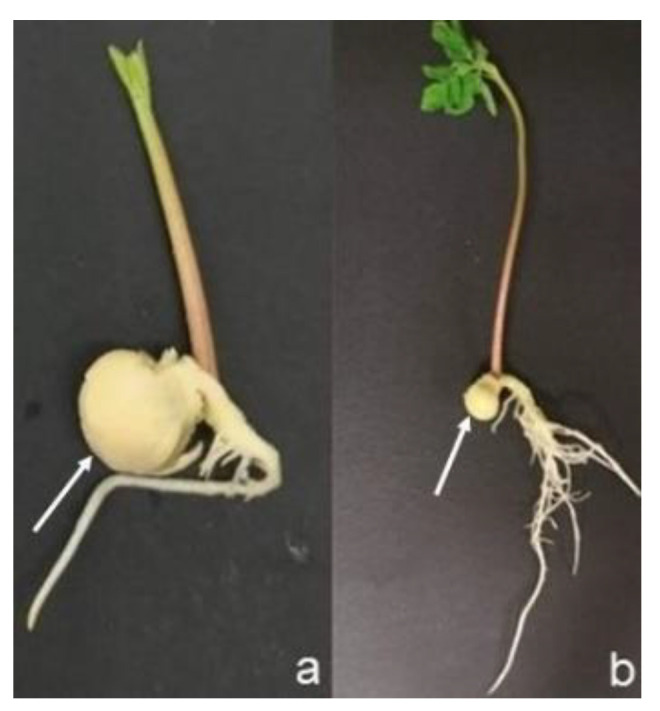
Analyzed Moringa sprout and cotyledon (arrows) developmental stages: (**a**) sprout with the first true leaves (Stage 1); (**b**) sprout with four leaves (Stage 2).

**Table 1 molecules-27-08774-t001:** Influence of sprout developmental stage and growing substrate on total phenolic contents (TPC), DPPH, ABTS and FRAP scavenging capacity of Moringa sprouts.

Developmental Stage	Growing Substrate	TPC		DPPH		ABTS		FRAP	
		mg GAE/g	±SD	mM TEAC	±SD	mM TEAC	±SD	mM TEAC	±SD
S1	Agarized medium	115.68	±6.41	190.90	±12.47	185.99	±5.76	134.56	±14.49
	Jiffy^®^	58.29	±0.66	48.37	±16.35	136.32	±11.87	7.72	±2.02
S2	Agarized medium	123.73	±16.43	148.89	±29.75	184.28	±23.95	72.47	±0.51
	Jiffy^®^	94.41	±17.37	114.85	±12.32	109.30	±5.98	78.25	±0.19
**Statistical analysis**		*p*		*p*		*p*		*p*	
Developmental Stage (DS)		0.065		0.416		0.220		0.023	
Growing Substrate (GS)		0.008		0.003		0.003		0.000	
DS × GS		0.184		0.016		0.270		0.000	

**Two-way ANOVA, Tukey’s test (*p*
< 0.05). Stage 1 (S1):** sprouts with two leaves; **Stage 2 (S2):** sprouts with four leaves; **Agarized medium:** HF culture medium; **Jiffy^®^:** peat pellets.

**Table 2 molecules-27-08774-t002:** Influence of sprout developmental stage and growing substrate on total phenolic contents (TPC), DPPH, ABTS and FRAP scavenging capacity of Moringa cotyledons.

Developmental Stage	Growing Substrate	TPC		DPPH		ABTS		FRAP	
		mg GAE/g	±SD	mM TEAC	±SD	mM TEAC	±SD	mM TEAC	±SD
S1	Agarized medium	3.27	±0.21	25.92	±2.22	25.95	±6.58	11.61	±0.25
	Jiffy^®^	7.91	±0.33	7.02	±1.60	41.28	±2.27	2.22	±0.78
S2	Agarized medium	5.32	±0.43	23.52	±0.82	38.06	±5.84	8.96	±0.18
	Jiffy^®^	12.04	±0.01	36.12	±1.55	41.48	±2.90	14.36	±0.36
**Statistical analysis**		*p*		*p*		*p*		*p*	
Developmental Stage (DS)		0.065		0.416		0.220		0.023	
Growing Substrate (GS)		0.008		0.003		0.003		0.000	
DS × GS		0.184		0.016		0.270		0.000	

**Two-way ANOVA, Tukey’s test** (***p*< 0.05**). **Stage 1** (**S1**)**:** sprouts with two leaves; **Stage 2** (**S2**)**:** sprouts with four leaves; **Agarized medium:** HF culture medium; **Jiffy^®^:** peat pellets.

**Table 3 molecules-27-08774-t003:** Influence of developmental stage and culture medium composition on total phenolic contents (TPC), DPPH, ABTS and FRAP scavenging capacity of Moringa sprouts.

Developmental Stage	Culture Medium Composition	TPC		DPPH		ABTS		FRAP	
		mg GAE/g	±DS	mM TEAC	±DS	mM TEAC	±DS	mM TEAC	±DS
S1	0.0 G	115.68	±6.41	190.90	±12.47	185.99	±5.76	134.56	±2.61
	0.5 G	71.69	±2.03	113.19	±27.66	206.48	±3.62	63.39	±20.27
	1.0 G	72.28	±0.85	124.63	±7.31	224.98	±9.26	75.76	±2.79
S2	0.0 G	123.73	±16.43	148.89	±29.75	184.28	±23.95	72.47	±0.51
	0.5 G	226.30	±78.60	320.19	±34.55	392.68	±5.08	305.53	±12.68
	1.0 G	143.30	±45.37	275.30	±22.35	352.18	±6.61	255.86	±6.62
**Statistical analysis**		*p*		*p*		*p*		*p*	
Developmental Stage (DS)		0.012		0.000		0.000		0.000	
Culture Medium Composition (CMC)		0.349		0.086		0.000		0.000	
DS × CMC		0.086		0.001		0.000		0.000	

**Two-way ANOVA, Tukey’s test** (***p*< 0.05**). **Stage 1** (**S1**)**:** sprouts with two leaves; **Stage 2** (**S2**)**:** sprouts with four leaves; **0.0 G:** HF medium; **0.5 G**: HF added with 0.5 mg/L of GA_3_; **1.0 G:** HF added with 1.0 mg/L of GA_3_.

**Table 4 molecules-27-08774-t004:** Influence of developmental stage and culture medium composition on total phenolic contents (TPC), DPPH, ABTS and FRAP scavenging capacity of Moringa cotyledons.

Developmental Stage	Culture Medium Composition	TPC		DPPH		ABTS		FRAP	
		mg GAE/g	±DS	mM TEAC	±DS	mM TEAC	±DS	mM TEAC	±DS
S1	0.0 G	3.27	±0.21	25.92	±2.22	25.95	±6.58	11.61	±0.25
	0.5 G	2.32	±0.04	4.56	±0.00	28.02	±3.48	<LOQ	<LOQ
	1.0 G	3.30	±0.40	4.95	±1.39	25.17	±5.21	<LOQ	<LOQ
S2	0.0 G	5.32	±0.43	23.52	±0.82	38.06	±5.84	8.96	±0.18
	0.5 G	8.28	±0.44	24.19	±3.76	40.92	±3.06	9.22	±0.18
	1.0 G	7.77	±0.62	27.32	±1.10	34.36	±4.71	13.98	±0.26
**Statistical analysis**		*p*		*p*		*p*		*p*	
Developmental Stage (DS)		0.000		0.000		0.003		0.000	
Culture Medium Composition (CMC)		0.002		0.000		0.436		0.000	
DS × CMC		0.000		0.000		0.853		0.000	

**Two-way ANOVA, Tukey’s test (*p*
< 0.05). Stage 1 (S1):** cotyledons from sprouts with two leaves; **Stage 2 (S2):** cotyledons from sprouts with four leaves; **0.0 G:** HF medium; **0.5 G:** HF added with 0.5 mg/L of GA_3_; **1.0 G:** HF added with 1.0 mg/L of GA_3_.

## Data Availability

Not applicable.

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
