# Peer review of "Sprouts of Moringa oleifera Lam.: Germination, Polyphenol Content and Antioxidant Activity"

_molecules, 2022, doi:10.3390/molecules27248774_

Round 1

Reviewer 1 Report

In the present study entitledSprouts of Moringa oleifera Lam.: germination, polyphenol con-
tent and antioxidant activity,” the Authors evaluated in vivo and in vitro, the germination of Moringa seeds and determined the polyphenol content and antioxidant activity of sprouts and cotyledons. The experiments and the results sound interesting and have high relevance.

I advise minor changes:

Please add more details to Figure 1, at least the abbreviations of the treatments.

Instead of data not shown, please add data as supplementary information to the manuscript.

Please, give the full name of the abbreviations at first mention in the text instead of the material and methods since it is at the end of the manuscript.

1mgl/l of GA3 in table 4, punctuation error in line 20, and many tiny typing errors or missing spaces can be found in the manuscript.

The resolution of Figures 1 and 4 is low. Please replace these figures with high-resolution ones.

Author Response

Please add more details to Figure 1, at least the abbreviations of the treatments. More details were added to Figure 1, including the legend of the thesis tested.

Instead of data not shown, please add data as supplementary information to the manuscript.

“Data nor shown” has been removed and a new graphic was added as supplementary material.

Please, give the full name of the abbreviations at first mention in the text instead of the material and methods since it is at the end of the manuscript.

Full name of germinative parameters was added in the Results paragraph.

1mgl/l of GA3 in table 4, punctuation error in line 20, and many tiny typing errors or missing spaces can be found in the manuscript.

Punctuation and typing errors were corrected.

The resolution of Figures 1 and 4 is low. Please replace these figures with high-resolution ones.
Resolution of figures was improved and figures were replaced.

Other than the above suggested revisions, Authors have checked the manuscript to improve the English language.

Reviewer 2 Report

Major comments:

Results section: Please write down what the acronyms stand for at the first instance they were described. What is FEP ? MET ? GV ? Although they were described in the Methods, the Methods was the last section. The readers will have to check the last section to find out what FEP or MET means.

Figure 1: What do 0.0G, 0.5G, 1.0G, and JIFFY stand for ? They should be written in the figure description. The Figures and Tables should stand alone; that is, the readers will know already what the data mean just by looking at the Figures and Tables, and without referring to the rest of the manuscript. Same comments with Figure 4. Additionally, what does "dd" on the right axis mean ?

Discussion section:

I find that the Discussion section lacks depth. There are not deep explanations on why your Results turned out like that. For example, the cotyledons had higher antioxidant content/activity when the seeds were germinated in Jiffy than in agarized medium. When you compared these results to those of the sprouts, the sprouts had higher antioxidant content/activity when grown in agarized medium than in Jiffy. Why were the trends different ? Would that suggest that more antioxidant metabolite precursors are retained in the cotyledon in the Jiffy-grown seedlings that is why the antioxidant level in the sprouts was lower ?

In terms of GA for sprouts, why was the general trend for antioxidants like this: 0.5G > 1.0G > 0.0G (if there was a significant change), and not like, 1.0G > 0.5G > 0.0G, as we usually would expect with the increase in GA? What could explain this trend ? Same comments for the cotyledons.

Table 4 needs adjustments - the columns are not aligned.

Author Response

Please write down what the acronyms stand for at the first instance they were described. What is FEP? MET? GV? Although they were described in the Methods, the Methods was the last section. The readers will have to check the last section to find out what FEP or MET means.
Full name of germinative parameters was added in the Results paragraph, at least at the first mention in the text.

Figure 1: What do 0.0G, 0.5G, 1.0G, and JIFFY stand for? They should be written in the figure description. The Figures and Tables should stand alone; that is, the readers will know already what the data mean just by looking at the Figures and Tables, and without referring to the rest of the manuscript.
Legend of different thesis tested was added, trying to better describe all the parameters reported in the figures.

Same comments with Figure 4. Additionally, what does "dd" on the right axis mean?
Legend of thesis was added and the “dd” was changed with “days”

I find that the Discussion section lacks depth. There are not deep explanations on why your Results turned out like that. For example, the cotyledons had higher antioxidant content/activity when the seeds were germinated in Jiffy than in agarized medium. When you compared these results to those of the sprouts, the sprouts had higher antioxidant content/activity when grown in agarized medium than in Jiffy. Why were the trends different ? Would that suggest that more antioxidant metabolite precursors are retained in the cotyledon in the Jiffy-grown seedlings that is why the antioxidant level in the sprouts was lower?
Secondary metabolism of plant may be influenced by several factors, as the culture conditions, temperature, type of substrate, etc. as evidenced in literature. It can be, thus, hypothesized that the different TPC and/or antioxidant content may vary on the basis of the type of explant (sprouts and cotyledons) and the cultural conditions adopted. We added a deepen discussion in lines 359-366.

In terms of GA for sprouts, why was the general trend for antioxidants like this: 0.5G > 1.0G > 0.0G (if there was a significant change), and not like, 1.0G > 0.5G > 0.0G, as we usually would expect with the increase in GA? What could explain this trend ? Same comments for the cotyledons.
The effect of GA3 addition was already investigated in literature. Different studies demonstrated that GA3 may induce the synthesis and the consequent increase of some secondary metabolites. We added this information in the introduction section (lines 86-92). Moreover, what the statistical analysis evidenced in this study is that the mere presence of GA3 stimulates the synthesis of secondary metabolites, independently of its concentration in the culture medium.

Table 4 needs adjustments - the columns are not aligned.
Tables were corrected.
Other than the above suggested revisions, Authors have checked the manuscript to improve the English language.

Round 2

Reviewer 2 Report

Much improved text !

Line 61: habit

Line 81: native to India

Line 87: are rich in nutrients

Line 150: Over the last decades, the research community

Line 151: emphasizing

Line 163: confirms the importance of using Moringa

Line 171: ... and other Far-Eastern countries, they are being introduced into the diets worldwide.

Line 183: Unfortunately, like many tropical seeds

Lines 183-184: Moringa seeds lose viability quickly, potentially due to their high metabolic rate

Line 189: respond well to 

Lines 192-193: independence from

Line 200: gibberellic acid (GA3

Line 236: triggering secondary metabolism

Line 237: may result in

Line 241: modulate secondary metabolism

Line 247: agarized medium

Lines 263-264: took to germinate, showed that on average, seeds germinated after 14.6 days.

Line 265: the calculated Germination value (GV)

Line 266: when sown in vivo

Line 270: Figure 1 title should be placed below the graph. Conventionally, figure titles are placed below the figure, and table titles above the tables.

Figure 1 - what does HF mean ?

Line 292: Moringa seed germination process.... Seeds swelled promptly

Line 293: and the protrusion of the first radicle

Line 294: around 33% of cultured seeds germinated

Lines 296-297: in 9 days, around 90% of sprouts had...

Line 329: conditions were seen

Line 332: in the culture medium was also highlighted

Lines 336-337: in the culture medium did not influence the Mean Germination Time (MGT).

Figure 4 title should be placed below the figure. Include also the meaning of 0.0G, 0.5G, and 1.0G as in Figure 1.

Line 345: confirmed that the addition of GA3 in the culture medium was ineffective

Line 347: was 56.9%

Line 394: 2.74 seeds per day

Line 398: showed FGP higher

Line 402: culture techniques for large-scale sprout production

Line 423: predominant effect resulting from the "Type of Sowing Substrate"

Line 424: in vitro sprouts were characterized

Line 427: with ABTS assay appeared to be exclusively determined

Line 430: with the results of the FRAP assay

Lines 433-434: On the other hand, in in vivo, the behavior was opposite

Line 469: Together with the sprouts, the cotyledons were also chemically analyzed

Line 474: those observed in cotyledons grown in vitro 

Line 476: stage 2 cotyledons showed

Line 484: the main source of variation

Line 485: cultured in vitro

Line 489: the ones grown in Jiffy

Line 492: if the substrate was considered, it appeared that extracts from stage 2 cotyledons, grown in Jiffy had a higher

Line 499: Moringa in vitro-cultured sprouts exhibited

Line 502: measured using the Folin-Ciocalteau assay

Line 504: resulted on average in richer polyphenols

Line 506: regardless of the culture medium composition

Line 507: FRAP assays showed a significant

Line 517: from in vitro-cultured sprouts

Line 521: medium appeared to increase

Line 523: activity of extracts was the "Sprout Developmental Stage"

Line 531: its multifaceted

Lines 533-534: Actually Leaves, flowers

Line 535: Moringa parts most widely studied, and they have been

Line 539: The well-documented health-promoting potential

Line 540: Moringa led to an increase in its cultivation

Line 546: plant biotechnology

Line 547: Actually, Several studies

Lines 550-552: This dearth of data made it difficult to perform comparative discussion of our results

Line 554: is utterly comparable

Line 563: seed germination. In vitro

Line 568: in vivo, and 6.4 days

Line 585: to deepen the understanding

Line 587: few are studies

Line 604: organs considered

Line 605: Actually, The TPC

Line 610: are subjected to during cultivation

Line 615: analyzed seems to have a crucial effect

Line 617, 618, 623: analyzed

Line 622: regardless of the extraction conditions

Line 624: is encouraging

Line 625: and after considering the limited effects

Line 630: In more detail

Line 681: is also reported

Line 690: are likely

Line 692: syntheses

Line 696: synthesize (Comment: analyse vs. analyze, for example, "analyse" is British English spelling and "analyze" is American English spelling). Check the rest of the manuscript for similar cases: characterise vs. characterize; snythesise vs. snythesize; behaviour vs. behavior; colour vs. color; characterisation vs. characterization, etc...

Line 701: activating defense mechanisms

Line 705: Echinacea purpurea

Comment: Check the entire manuscript - should be "in Jiffy" not "on Jiffy"

Line 908: cultures... dietary habits worldwide

Line 911: that the germination process

Line 958: a long history

Lines 959-960: source of medications... it is getting attention for its application in the pharmaceutical

Line 978: in vitro, and in their cotyledons

Comment: Maybe include in the Conclusion a short outlook on how in vitro cultivation can be adopted and what are the limitations of in vitro cultivation (costs ? scalability ?)
